# Polysulfone Influence on Au Selective Adsorbent Imprinted Membrane Synthesis with Sulfonated Polyeugenol as Functional Polymer

**DOI:** 10.3390/membranes10120390

**Published:** 2020-12-03

**Authors:** Muhammad Cholid Djunaidi, Nor Basid Adiwibawa Prasetya, Arini Khoiriyah, Pardoyo Pardoyo, Abdul Haris, Nabilah Anindita Febriola

**Affiliations:** Department of Chemistry, Faculty of Science and Mathematics, Diponegoro University, Semarang 50275, Indonesia; nor.basid.prasetya@live.undip.ac.id (N.B.A.P.); arinikhoiriyah@students.undip.ac.id (A.K.); pardoyoku@live.undip.ac.id (P.P.); a.haris@live.undip.ac.id (A.H.); nabilahanindita@students.undip.ac.id (N.A.F.)

**Keywords:** imprinted membrane, sulfonated polyeugenol, polysulfone, gold adsorption, selectivity

## Abstract

An ionic imprinted membrane (IIM) was synthesized using sulfonated polyeugenol, derived from eugenol, as its functional polymer and polysulfone as its base membrane for the selective adsorption of Au(III). This study aims to determine the adsorption of Au(III) metal ions using IIM compared with the non-imprinted membrane (NIM) and to figure out the membrane selectivity towards Au(III) in mixed solutions of Au/Cd, Au/Cu, and Au/Fe. IIM has a pore size of 0.767 μm while the non-imprinted membrane (NIM) has a pore size of 0.853 μm. The best adsorption result was obtained in the variation of the membrane with the addition of 3.84 g of polysulfone that had pores according to the size of Au. The selectivity results of the Au/Cd mixture solution in NIM and IIM were 17.802 and 36.265. In the mixture of Au/Cu, the NIM and IIM selectivity was 2.386 and 6.886, and in the mixed solution of Au/Fe, the selectivity of NIM and IIM was 0 and 8.489. Thus, the selectivity of IIM towards Au is bigger than NIM.

## 1. Introduction

Precious metals such as gold (Au) have high economic value and possess various characteristics compared to other metals. Gold can conduct electricity well, malleable, and can be used in various kinds of ornaments such as coins, jewelry, sculpture, electronic products, and catalysts [1]. As technology develops, gold becomes more broadly applied in industrial process due to its catalytic [2] and corrosion-resistant properties [3]. Industrial processes often leave waste containing useful and valuable metals such as gold [4]. So, waste treatment must be carried out to extract this valuable metal that later on could be used again in the industrial process. Gold concentration in an aquatic environment is low enough that gold determination and recovery require several pre-treatment methods. Due to the low concentration of gold ions and the presence of pre-concentration interference, selective metal separation is necessary [5]. There are several techniques for gold recovery such as chemical deposition, ion exchange, adsorption, and membrane filtration [6].

A previous study that has been done regarding Au recovery is by Yuliani who synthesized an ionic imprinted membrane (IIM) which was selective for Au(III) ions using membrane transport techniques, using polyeugenoxy acetyl thiophene methanolate (PEATM) as a functional polymer. The study showed that the transport in a binary mixed metal solution using IIM-PEATM was more selective for transporting Au(III) metal when compared to NIM PEATM. This proves that the presence of the SO_3_^−^ group in PEATM makes IIM and NIM membranes more selective against Au(III) metal ions which are per the HSAB (Hard Soft Acid Base) theory and it also proves the presence of imprinted ion template in IIM that makes the membranes more selective towards Au(III) [7]. Also, Firlak conducted a study on the recovery of Au(III) ions with imprinted Au(III) hydrogel in water samples. The results of this study indicate that the hydrogel template can selectively adsorb Au(III) ions for high and low concentrations of gold [8].

In this study, sulfonated polyeugenol derived from eugenol will be used as the functional polymer for the synthesis of IIM for Au adsorption. Eugenol is one of the materials that can be used for the separation of metal ions. Eugenol is the starting material for the synthesis of a new compound such as polyeugenol, due to the presence of three functional groups attached to it, namely the allyl, hydroxyl, and methoxy groups, making it a potential functional polymer in synthesis of IIM for selective adsorption [9]. Eugenol is also easily found in Indonesia, it is found in essential oils produced from clove, a very abundant plant in Indonesia. IIM and NIM were synthesized using eugenol derivatives in the form of sulfonated polyeugenol for the adsorption of Au(III) ions. Then, binary metal ions such as Cd, Fe, and Cu were used to determine the membrane selectivity to Au metal. This study will also explain the importance of functional groups in the selective adsorbent membrane. So, it is necessary to determine Au(III) selectivity by IIM adsorption to determine the functional polymer effect towards Au selectivity, more specifically, the sulfate group effect in the functional polymer.

## 2. Materials and Methods 

### 2.1. Materials

Eugenol p.a, BF_3_-diethyl ether, methanol, anhydrous Na_2_SO_4_, AIBN (2,2’, Azobis(2-methylpropionitrile)), polysulfone and polyethylene glycol were purchased from Sigma-Aldrich (Jakarta, Indonesia). 0.1 M HCl, H_2_SO_4_, HNO_3_, chloroform p.a, 1000 ppm Au solution, NMP (1-Methyl-2-pyrrolidone), thiourea, and metals ions in the form of Cd(NO_3_)_2_, Cu(NO_3_)_2_, and Fe(NO_3_)_2_ were purchased from Merck (Jakarta, Indonesia) while aquabidest was purchased from Bratachem (Semarang, Indonesia).

### 2.2. Polymerization of Eugenol

In a three-necked flask, 5.8 g of eugenol was polymerized using BF_3_-diethyl ether as a catalyst under stirring for 16 h, the polymerization was then ended by adding methanol. The polymerization result was dissolved with chloroform and neutralized by adding water. Afterward, anhydrous Na_2_SO_4_ was used to remove the water completely and then filtered. Then, it was let stand for 24 h until the chloroform solvent evaporates. Afterward, polyeugenol was crushed until a pink powder is obtained.

### 2.3. Synthesis of Sulfonated Polyeugenol

Sulfonated polyeugenol (SPE) was synthesized by sulfonating 0.5 g of polyeugenol using 7.5 mL of H_2_SO_4_ 99% and 0.05 g of PbSO_4_ by heating using reflux for 15 min, then leaving it at room temperature to cool down. Later on, 50 mL of H_2_SO_4_ 2M (cold) was added and then filtered and neutralized with distilled water. The polymer was then analyzed by FTIR. Furthermore, the SPE was contacted with Au(III) at pH 3 condition with a composition ratio of 1:20 as in 1 g of SPE and 20 mL of Au(III), and the results of the contact (SPE-Au(III))were analyzed using AAS (PerkinElmer Inc., Waltham, MA, USA) and XRD (Shimadzu Coporation, Kyoto, Japan).

### 2.4. Molecular Weight Measurement of Polymer

Polyeugenol was dissolved in NMP to a concentration of 0.002 g/mL, 0.004 g/mL, 0.006 g/mL, 0.008 g/mL, and 0.010 g/mL. Then, the pure solvent flow time (NMP) as t_0_ and the polymer flow time from the lowest to the highest concentrations as t_1_, t_2_, t_3_, t_4_, and t_5_ was measured using the Ubbelohde Viscometer. Through calculation, the specific viscosity (ƞ_sp_) will be obtained using the Huggins equation which is derived to be a linear regression equation (y = mx + c), where the *y*-axis is the reduced viscosity value (ƞ_sp_/C) and the *x*-axis is the concentration value (C). In the graph, the resulting intercept value (c) is the intrinsic viscosity value [ƞ]. With the Mark–Houwink–Sakurada equation, [ƞ] = KMva, molecular weight (Mw) can be calculated with the value of K = 11 × 10^−3^ and a = 0.725. The same method was also used to measure sulfonated polyeugenol molecular weight.

### 2.5. Synthesis of Imprinted Membrane

The synthesis of NIM used polysulfone (PSf) variations of 3.34 g, 3.59 g, and 3.84 g. Then, each was added with 0.833 g of polyethylene glycol (PEG), 0.833 g of SPE, 12 mL of NMP, and 0.015 mL of AIBN catalyst to each variation. The mixture was then refluxed for 10 h at a temperature of 90–100 ℃. IIM was synthesized with the same method but uses SPE-Au(III) instead of SPE. The results of the synthesis of the NIM and IIM were let stand for 24 h, then the membrane mixture was casted on a glass surface with the membrane thickness adjusted to be 0.05 mm–0.08 mm and then immediately immersed into aquabidest. Next, the membrane was washed using a solution of 0.8 M of thiourea in 0.1 M of HCl for 2–3 weeks. After washing the membrane, it was neutralized with distilled water. The membranes were analyzed by FTIR (Shimadzu Corporation, Kyoto, Japan) and SEM-EDX (JEOL Ltd., Tokyo, Japan).

### 2.6. Membrane Adsorption on Au(III) Metal Ions and Selectivity Test

Adsorption of 25 ppm of Au(III) solution at pH 3 conditions by NIM and IIM membranes were conducted using a shaker for 6 h and sampling was done every 1.5 h to find the optimum time. The adsorption results were analyzed using AAS. For the selectivity test, The NIM and IIM membranes in Erlenmeyer were added with a mixture of Au/Cd, Au/Cu, and Au/Fe solutions with a concentration of 10 ppm. Then, using a shaker, the adsorption was done for 270 min and the results were analyzed using AAS. 

## 3. Results

### 3.1. Eugenol Polymerization

Eugenol polymerization was carried out for 16 h and was stopped with methanol to give the methoxy group at the end of the polymer chain. Then, the solution was neutralized using distilled water and filtered with anhydrous Na_2_SO_4_ to bind the water contained in the polyeugenol. Eugenol polymerization process yielded 5.20 g (89.65%) of polyeugenol with a molecular weight of 6225.5 g/mol. The result is a pink powder that is soluble in methanol and chloroform. The polymerization process has three stages:

Initiation Stage: 

The addition of BF_3_-diethyl ether as a Lewis acid catalyst, due to its easily reduced nature, will cause the protons (H^+^) in it to be transferred to the eugenol monomer and an addition reaction will occurs at the initiation stage, causing the allyl group on eugenol to break the double bond. This is because the empty orbitals in BF_3_ are transferred to the eugenol monomer to form carbonium ions. This reaction is characterized by the solution color that changes into a purplish red color. This addition reaction follows Markovnikov’s law, where the stability of the carbonium ion determines the reactivity for the subsequent monomer incorporation.

Propagation Stage:

In the propagation stage, the formation of eugenol monomer chain is characterized by the formation of covalent bonds between the cation chains and the eugenol monomers. This process is continued until a long polymer chain is obtained. At this stage, intermolecular rearrangement of the carbonium ion also occurs.

Termination Stage:

The cationic polymerization reaction of eugenol with BF_3_-diethyl ether catalyst will take place until all the monomers are used up. At this stage, a fusion reaction between the propagated chain (carbonium ion chain) and the monomer occured. In the end, methanol was added to stop the polymerization. The reaction is as seen in Figure 1:

Eugenol and polyeugenol that has been synthesized were analyzed using FTIR, the result can be seen in Figure 2 below:

In polyeugenol, there is an absorption band of 2958.48 cm^−1^ that indicates the C sp^2^-H absorption band from the aromatic ringand an absorption band at 1512.18 cm^−1^ and 1603.17 cm^−1^ which shows the vibration type of the C=C aromatic range, this strengthens the proof of aromatic groups presence in polyeugenol. The absorption band in the area of 2958.48 cm^−1^ and also strengthened by the absorption band at 1458.5 cm^−1^ showed the characteristic of C−H stretch from the methyl group. Before polymerization, eugenol has a vinyl group on the 995 cm^−1^ and 910 cm^−1^ absorption bands, but there were none in polyeugenol. This happens because the vinyl group has been modified so that it binds to other eugenols to form polyeugenol. Then, an absorption band at 817 cm^−1^ appeared which indicated the presence of ortho-substituted aromatic compounds, the −OCH_3_ group and an absorption band at 756 cm^−1^ which indicated the presence of para-substituted aromatic compounds, the CH_2_=CH−CH_2_− groups. This proved the presence of aromatic compounds in eugenol and polyeugenol.

### 3.2. Sulfonated Polyeugenol Synthesis 

Polyeugenol sulfonation was carried out using concentrated H_2_SO_4_ and yielded 0.4 g (80%) of sulfonated polyeugenol with a molecular weight of 7742.3 g/mol. It is hoped that the sulfonation treatment of polyeugenol can change the properties of the polysulfone membrane from hydrophobic to hydrophilic. The addition of SO_3_H groups is very suitable for Au(III) adsorption based on the HSAB (Hard Soft Acid Base) principle. The sulfonic acid group will enter the meta position respective to the −OMe group because the −OH group are both ortho and para director. The ortho position in the right and the para position were already filled, so the SO_3_^−^ group will enter the left ortho position. The reactions that occur in the polyeugenol sulfonation process are seen in Figure 3:

The results of FTIR analysis on the sulfonation of polyeugenol are shown in Figure 4:

### 3.3. Synthesis of NIM and IIM Results

Membranes were synthesized from PSf and SPE/SPE-Au(III) that were cross-linked with PEG which has −OH groups at each end. One of the OH group ends will bind to the polysulfone in benzene which is close to the −O− group and the O=S=O group, this is because the −O− group is an ortho director and is supported by the O=S=O group which is a meta director. So, the −OH at one end of the PEG will bind to the benzene ring on the polysulfone while the other −OH end of the PEG binds to the functional polymer, SPE-Au(III) at the polymer −OH end. Figure 5 presented the result of the crosslink reaction that is expected to occur in the membrane:

XRD analysis was also done to prove Au(III) presence in the polymer. The sample XRD spectra was compared to the standard Au(III) spectra achieved from RRUFF number of R070279. In the comparison of the sample and standard spectra, four similar peaks are obtained which indicates that there is Au(III) in the sample. The spectra result can be seen in Figure 6.

The reaction that occurs in a non-imprinted membrane or NIM is the same as the reaction in IIM-Au(III), except that there is no template of Au(III) in the membrane. The crosslink reaction in the membrane may also lock the Au(III) template on the SPE polymer so that a suitable imprinted mold could be formed. The resulted membrane was then analyzed by FTIR and the result can be seen in Figure 7 below.

In polysulfone FTIR result, there is a S=O group absorption band at 1200–1300 cm^−1^ and also the absorption in the range of 1230–1236 cm^−1^ which indicates that the intensity of the polysulfone S=O group vibration increased. So, it may be concluded that the polysulfone in the membrane increased.

Characterization using SEM-EDX was done to determine the morphology, pore size, and composition of elements contained in NIM and IIM. The SEM analysis results were then processed further using image-J software to sharpen the resulting pores. The results can be seen in Figure 8.

The EDX results showed the composition of the elements contained in NIM and IIM and are shown in Table 1.

In the synthesis of NIM and IIM, the amount of PSf used were varied to 3.34; 3.59; and 3.84 g. This is intended to study the effect of PSf amount on the adsorption ability of NIM and IIM. In Figure 9, SEM characterization of the varied PSf membrane was done to determine the surface morphology and the pores of each membrane.

### 3.4. Adsorption of Au(III)

The synthesized membrane is used for the adsorption of Au(III) solution at pH 3 conditions with a concentration of 25 ppm. pH 3 is the optimal condition for the adsorption of Au(III) metal ions according to Yuliani’s research in 2019 where the pH optimization of the feed phase was carried out for the transport of Au(III) metal ions. The optimal results achieved were at pH 3 because the percentage of remaining metal ion concentrations in the receiving phase was the least when compared to pH 1, 5, 7, and 9. The reaction that occurs in the receiving phase is influenced by the pH conditions of the feed phase, where the reduction reaction of Au(III) ion to Au(I) occurs in an acidic conditions (in the feed phase) [10]. So, if the condition of the Au(III) ion from the receiving phase is also acidic, it will make the reduction reaction run faster, this is due to the driving force transport of Au(III) ions that increased and will decrease with increasing pH [8]. Also, in Paclawski and Fitzner (2004) study about the distribution of Au(III) species at various pH conditions, at a pH of more than 3, the presence of hydroxyl ligands begins to replace Cl ligands so that at pH above 3 there will be [AuCl_3_OH]^−^, [AuCl_2_(OH)_2_]^−^, [AuCl(OH)_3_]^−^, and [Au(OH)_4_]^−^ species. The existence of different species of Au(III) metal ions is likely to decrease the adsorption [11]. The adsorption results using NIM and IIM with PSf amount variations are shown in Table 2:

From the AAS results, it was concluded that the more PSf was added, the pores become more uniform and smaller [12], adjusting to the size of Au(III) particles, and the adsorption of Au(III) metal ions also increased. The trendline of each membrane adsorption results can be seen in Figure 10:

### 3.5. Selectivity Test Results

Adsorption selectivity test of the membranes that had been synthesized was carried out to determine the selectivity difference between the NIM and IIM. The mixture of metal ions used to determine IIM selectivity are a solution of Au/Cd, Au/Cu, and Au/Fe. Where according to the HSAB classification, the Cd^2+^ metal is a soft acid group, Cu^2+^ metal is a medium acid group, and Fe^3+^ is a hard acid group. So, the presence of competing metal ions can be used to determine the selectivity of NIM and IIM on the adsorption of Au(III) metal ions [13].

The binary metal solution mixture is at an optimum pH condition (pH 3) and in the concentration of 10 ppm. The selectivity study used Au concentration of 10 ppm due to the adsorption capacity so that the selectivity can be seen without exceeding its adsorption capacity. Adsorption was carried out for 4.5 h using a shaker, to determine the selectivity ratio between NIM and IIM. After the adsorption was carried out, the results of the solution were analyzed using AAS to determine the concentration of Au metal ions and competitor metal ions in the remaining solution mixture. The membrane selectivity resultsfor a mixture of Au/Cd solutions, Au/Cu solutions, and Au/Fe solutions are shown in Figure 11:

## 4. Discussion

### 4.1. Polysulfone and Au(III) Metal Ions Additions Effects on Membrane Pores 

The SEM characterization results are used to measure the pore size by Image-J software, the results showed that the porous membrane morphology with the average pore of NIM was 0.853 μm and IIM was 0.767 μm. From the morphological images of NIM and IIM, it was found that the pore size of IIM was smaller and more uniform, this could be due to the influence of the Au(III) metal ion printed on the membrane. The Au metal ion has a radius size of 275 nm in the state of the hydrated ion species Au(III) [9], causing the membrane to have more uniform pores by adjusting to the small size of Au(III).

Based on research conducted by Ficai (2010) [14], polysulfone can regulate the desired porosity if the desired particle is inserted, in that study the particles undergo agglomeration which causes the polysulfone membrane to gather and adjust to the size of the particles so that the pores on the PSf membrane will have almost the same size as the particles [14]. Whereas in this study, the particles used were metal ion Au(III) with a size of approximately 275 nm in the hydrated ion species state, an increase in polysulfone will produce a defect-free membrane with the desired pores (following the size of the Au(III) metal ion). The pores of the membrane with the addition of 3.34 g PSf were 1.561 μm, PSf addition of 3.59 g showed pores size of 1.366 μm, and PSf addition of 3.84 g resulted in pores size of 0.644 μm. Away (2013) analyzed the effect of PSf addition on the effect of selectivity. Membranes with a PSf concentration of 25% had the best selectivity levels compared to the PSf addition of 20%, 22.5%, 27.5%, and 30%. The PSf concentrations of 20% and 22.5% have low selectivity values because the composition of the base material is less than the solvent, causing the formation of polymer chains to be not optimal. The selectivity value decreased at 27.5% and 30% because the PSf concentration was too high, causing the selective layer to be so dense that it did not function properly [15].

### 4.2. Effect of Polysulfone Addition on Membrane Adsorption

From the NIM and IIM graphs of each variation, it was found that the graph with the addition of PSf of 3.84 g had the best trendline, meaning that the adsorption for 6 h increased on the IIM membrane and the NIM experienced instability due to fluctuating adsorption. This proves that the presence of imprinted Au(III) metal ions increases the IIM adsorption of Au(III). The best membrane adsorption was obtained at the addition of PSf of 3.84 g because the pores on the membrane were more uniform, smaller, and more similar the size of Au(III) metal ion, so that adsorption on Au(III) would be more optimal.

### 4.3. Influence of Membrane Template on Adsorption Selectivity

In the mixed solution of metal ion Au/Cd, the IIM selectivity value was greater than NIM. IIM selectivity value is 36.265 while the NIM selectivity value is 17.802. The membrane with a mixed solution of Au/Cd, showed pretty high Cd metal ion adsorption. Cd metal is a very strong competitor because both of Cd(II) and Au(III) belong to the same category according to the HSAB theory which is the soft acid group. Both of them could be drawn to the active site of the membrane that is a soft base group. However, IIM adsorption of Au metal ions still has the highest value due to the presence of a Au(III) metal ions template which makes it easier for Au(III) metal ions to be adsorbed into the membrane because in the membrane there is an S group which is a soft base group in HSAB that can attract Au(III) metal ions. Also, Cu metals that belong to the borderline acid group showed fairly low adsorption when compared to the adsorption of Au metal in IIM. The resulting selectivity also has a high enough ratio between NIM and IIM, with selectivity in NIM Au/Cu of 2.386 and IIM Au/Cu of 6.886. This is also due to the imprinting of the Au(III) metal ion in IIM. The last solution mixture is Au/Fe where Fe is a hard acid which has the least adsorption rate of IIM-Au(III), this is because the S group in the membrane is a soft base group and Fe metal ion is a hard acid group. The Fe metal ions will be difficult to adsorb into the membrane. Also, the radius of the hydrated ion Fe(III) is larger at 0.288 nm compared to Au(III) with the radius of the hydrated ion of 0.275 nm. So, in NIM, it can be seen that the membrane cannot adsorb Au(III), which is thought to be due to the pores. The pore is closed by Fe(III) and is unable to pass Au(III). The selectivity shown between NIM and IIM has a fairly large ratio, NIM Au/Fe has a selectivity value of 0 and IIM Au/Fe has a selectivity value of 8.489. Some source of experimental error in this study may be caused by the hot plate’s inability in maintaining the temperature perfectly during the IIM and NIM synthesis process, so the reflux process was conducted in the temperature ranging from 90 to 100 ℃. The results of the study have similarities to previous research conducted by Djunaidi in 2016, where membranes made through imprinting technology (molecular or ion imprinting) demonstrated higher selectivity than non-imprinted membranes [9].

## 5. Conclusions

Polyeugenol, a derivative compound of eugenol, has been successfully synthesized with 89.65% yield. Sulfonated polyeugenol has also been successfully synthesized with a yield of 80%. A non-imprinted membrane and ionic imprinted membrane have been successfully synthesized with an average thickness of around 0.05–0.7 mm. The effect of adding polysulfone to the membrane can reduce the diameter of the pores and also make the pores more uniform. The adsorption of Au(III) using IIM showed satisfying results as it is greater than NIM. This is due to the Au(III) template in the IIM. The selectivity of the ionic imprinted membrane to Au(III) metal ions in a mixture of binary metal ion solutions is greater than that of the non-imprinted membrane.

## Figures and Tables

**Figure 1 membranes-10-00390-f001:**
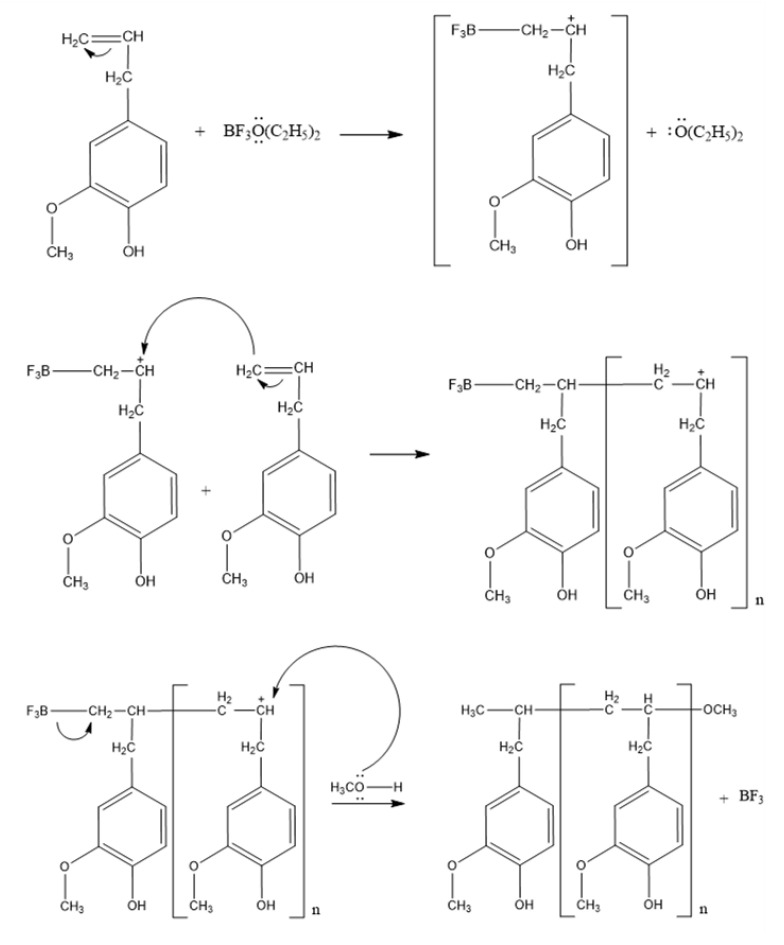
Polymerization process of eugenol.

**Figure 2 membranes-10-00390-f002:**
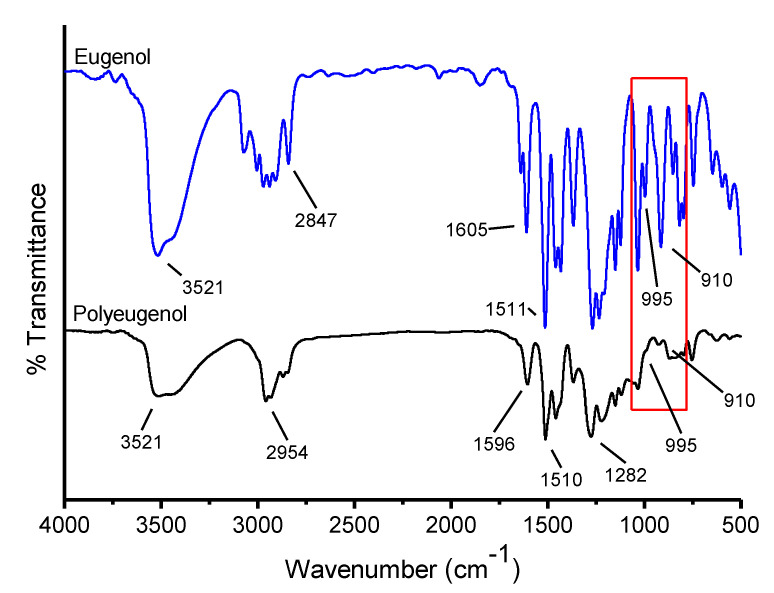
FTIR Graph of eugenol and polyeugenol.

**Figure 3 membranes-10-00390-f003:**
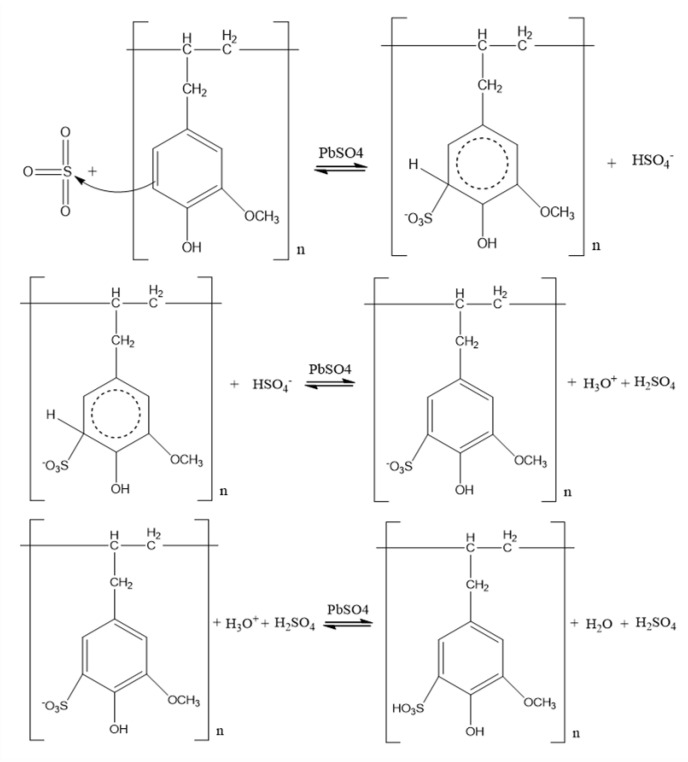
Sulfonation mechanism of polyeugenol.

**Figure 4 membranes-10-00390-f004:**
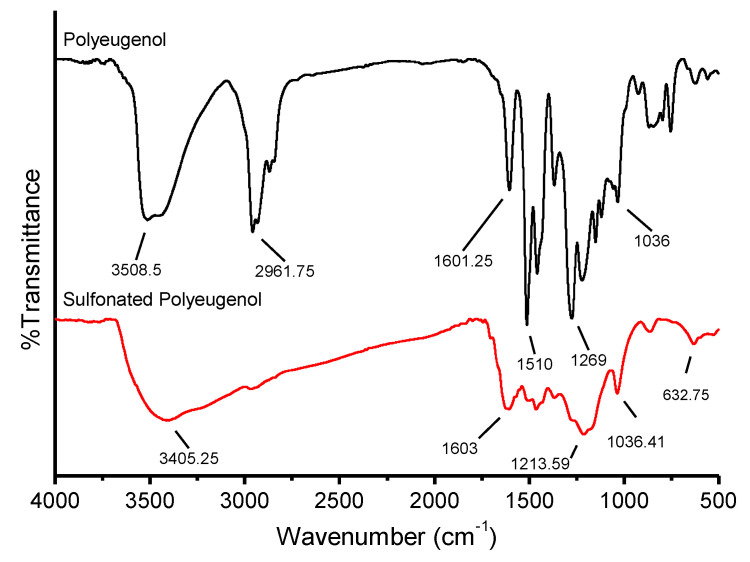
Graph of polyeugenol and sulfonated polyeugenol.

**Figure 5 membranes-10-00390-f005:**
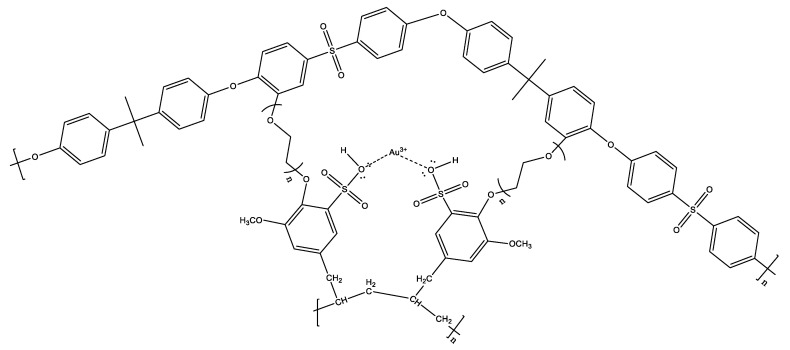
Estimation of crosslink bond in IIM-Au(III).

**Figure 6 membranes-10-00390-f006:**
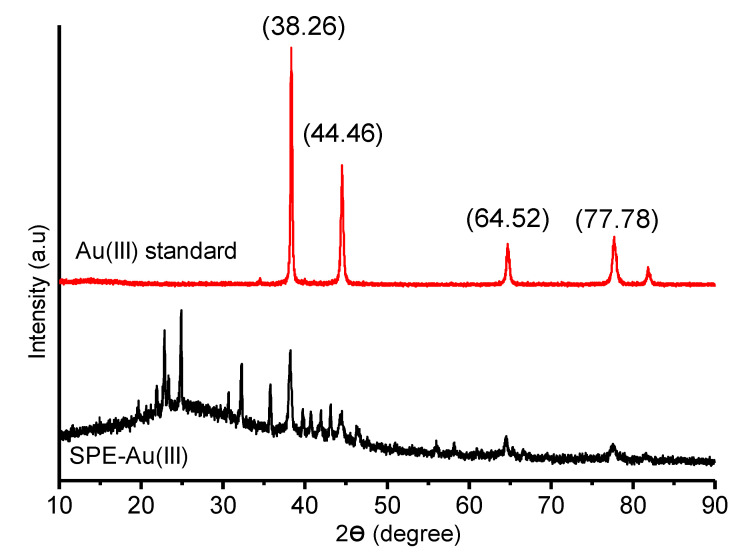
XRD analysis of SPE-Au(III).

**Figure 7 membranes-10-00390-f007:**
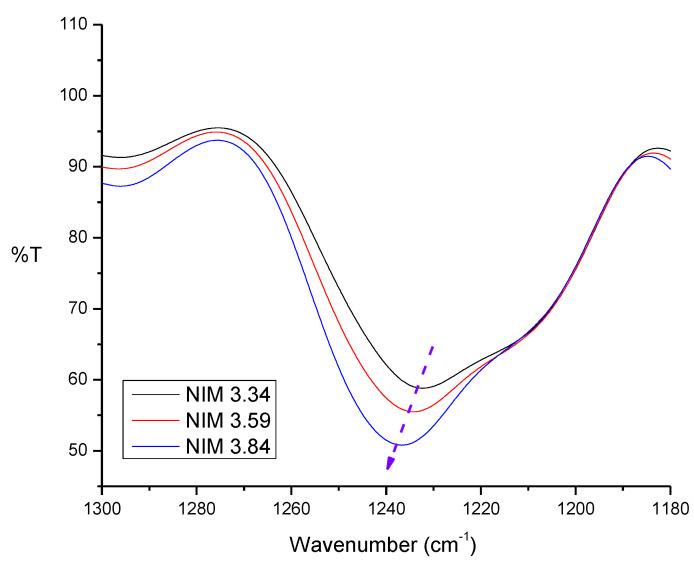
Comparison of membranes with variations of polysulfone addition.

**Figure 8 membranes-10-00390-f008:**
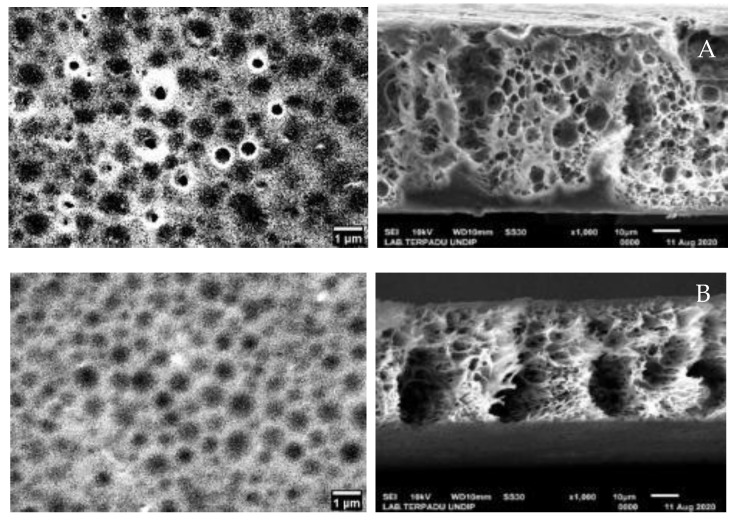
Surface morphology and cross-section before template release of (**A**) NIM (**B**) IIM.

**Figure 9 membranes-10-00390-f009:**
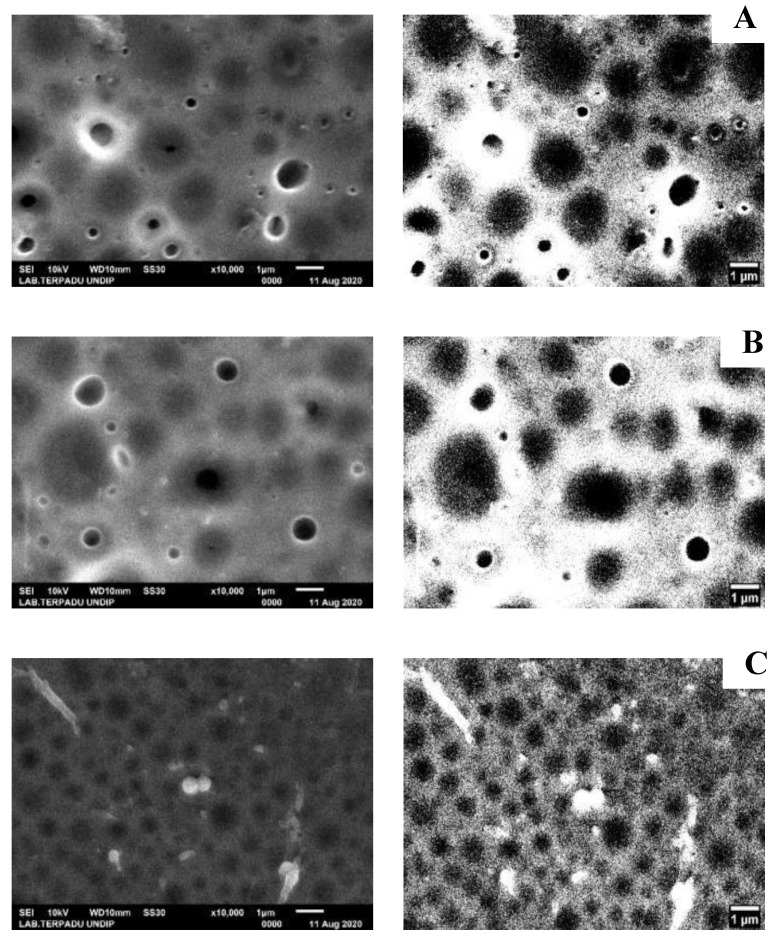
IIM surface morphology with the polysulfone addition of (**A**) 3.34 g (**B**) 3.59 g (**C**) 3.84 g.

**Figure 10 membranes-10-00390-f010:**
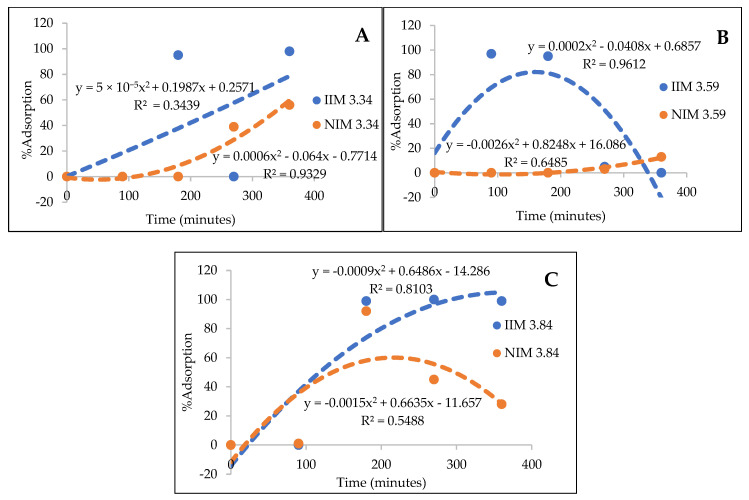
Adsorption result of NIM dan IIM with the polysulfone addition of (**A**) 3.34 g (**B**) 3.59 g (**C**) 3.84 g.

**Figure 11 membranes-10-00390-f011:**
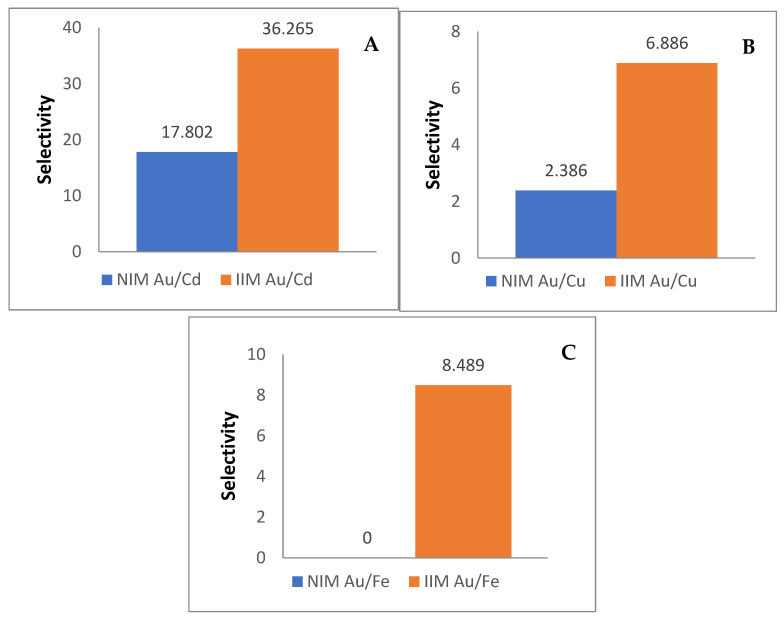
Selectivity of NIM and IIM in (**A**) Au/Cd (**B**) Au/Ce (**C**) Au/Fe mixture solution.

**Table 1 membranes-10-00390-t001:** EDX analysis results of the elements in NIM and IIM.

Elements	NIM Mass (%)	IIM Mass (%)
C	86.21	77.29
O	8.27	18.13
S	5.52	4.58

**Table 2 membranes-10-00390-t002:** Adsorption percentage of NIM and IIM with various PSf additions.

Adsorption Duration (Minutes)	3.34 g PSf		3.59 g PSf		3.84 g PSf	
	NIM	IIM	NIM	IIM	NIM	IIM
0	0%	0%	0%	0%	0%	0%
90	0%	0%	0%	97%	1%	0%
180	0%	95%	0%	95%	92%	99%
270	39%	0%	3%	5%	45%	100%
360	56%	98%	13%	0%	28%	99%

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
