# Peer review of "Polysulfone Influence on Au Selective Adsorbent Imprinted Membrane Synthesis with Sulfonated Polyeugenol as Functional Polymer"

_membranes, 2020, doi:10.3390/membranes10120390_

Round 1

Reviewer 1 Report

Manuscript title: “Synthesis of Ionic Imprinted Polysulfone Membrane for Au(III) Adsorption Based on Sulfonated Polyeugenol with Polyethylene Glycol As a Crosslinking Agent”

##Overall comments

The paper outlines the synthesis of gold ion-printed polysulfone membrane and the adsorption of gold ion from 25 ppm synthetic gold salt solution. I have found numerous drawbacks against publication in this current manuscript, such as paper presentation, IIM synthesis, and English. I, therefore, recommend a major revision.

Addressing the following issues will make this paper more impactful

## Comments on abstract, title, references

Title of the paper is not impactful, suggest revising the title.

Abstract of the paper is very general. Please include the pore size of IIM and NIM. Write the full form of Psf. Write the potential utility of this research work.

Lin 23: “It has more uniform pores and the size is similar to that of Au(III)” What do authors want to tell.

Reference must be updated and need to include more references.

## Comments on introduction/background

The introduction part is very poor and chaotic. Need rigorous revision.

Line 30-31: Sentence should be corrected. I don’t found any report regarding corrosion resistance study from Reference [2]. Please check and confirm.

Line 34 “Industrial processes often leave waste containing useful and valuable metals such as gold” Please include a reference.

Line 41-44: “Eugenol is one of the materials that can be used ……..” same sentences are used in the abstract. Please stop the repetition.

Line 45-46: Please revise this sentence because meaning is not clear.

What is the sense of reference of [6] inline 53-54?

Line 61: What is S group?

Line 67-68: Need revision.

Line 75: “So it is necessary to do an overall test of the sulfate functional group against the metal ion Au(III)” What do authors mean that?

## Comments on methodology

The methodology section should be carefully written. I have found several mistakes such as: See this sentence. The polymerization results were neutralized and filtered using Na2SO4. Then it was let stand for 24 hours until the chloroform solvent evaporates” What is the role of Na2SO4? if it is for water absorption then question is that wherefrom water will be generated in the reaction and Wherefrom the chloroform generated that need to evaporate?

Line 87: What was the concentration of sulfuric acid?

Line 88: “50 mL of H2SO4 2M (cold) was added, then filtered and neutralized with distilled water. I am interested to know about, is there any degradation of the polymer in the strong acidic medium?

Line 90-91: “Furthermore, the polyeugenol sulfonate was contacted with Au(III) at pH 3 condition with a composition ratio of 1:20, and the results of the contact (PES-Au(III)) were analyzed using AAS and XRD.” write details of composition ratio 1: 20 means which one is for which and ratio in wt % or volume. I don’t find XRD results, please include it.

Synthesis of the imprinted membrane which is the main story of this paper is not presented in detail. Please revised it carefully in details. If possible, include one schematic of the synthesis process.

Why authors have used 25 ppm of Au (III) solution? And why 10 ppm for selective study?

## Comments on data and results

Line 115: “The result is a pink powder that is soluble in methanol and chloroform. I can’t correlate this sentence because authors have written that methanol was used to end this polymerization reaction. How could the authors explain this sentence?

The mechanism should be represented in a Figure including (Initiation, propagation and termination).

Fig. 4 and Fig. 6: Peak value should be included for all major peaks.

Fig. 5: Why the sulfonic acid group enter to the m-position respective to –OMe group? Please explain it.

Fig. 7 is not clear to me, please check once again.

How do authors evaluate the pore size using SEM micrographs? Authors should study the BET or Gas permeability test to get a clear idea about the porosity of the membrane.

Authors have stated that due to HSAB principle Cd (II) and Au (III) ion absorbed nicely by Au-IIM. Please explain in details.

Line 260-262: please revise this sentence.   

## Comments on conclusions

How the authors measured molecular weight and reported in the conclusions is not clear. Another part has been presented according to the experimental results. Authors should write about the measurement of Mw.

Author Response

Dear Reviewer 

Reviewer 2 Report

This work can be published in Membranes if the following comments are addressed in the revised manuscript carefully.

  1. In abstract add significant contribution and outcome of present work.
  2. Needs proofread in the whole manuscript to improve language and mistakes
  3. Manuscript contains poor literature review. Authors should discuss more recent relevant works
  4. The novelty and necessity of this research need to be clearly mentioned in the introduction part.
  5. How would this research and its results be beneficial to academia and/or industry?
  6. Materials and method section contains so many small sub-section. It is suggested to reduce these subsections.
  7. Text writing is not consistent in the sentences, tables and figures

For example: “5.2 g” (in line 22), “86,21” (in Table 1) “3,34 g PSf” (in Table 2, etc),

“Figure 10. IIM surface morphology with the addition of (a)3.34 g PSf (b) 3.59 g PSf (c)3.84 g PSf”

“Figure 11. Adsorption result of NIM dan IIM with PSf addition of (a) 3,34 g (b) 3,59 g (c) 3,84 g”

8. What are the errors in the experimental data? The sources of error should be added to the revised manuscript.

Author Response

Dear Reviewer 2

Round 2

Reviewer 1 Report

I have revised the revision and found improvement. I recommend this paper for publication. Best wishes!